# Prevalence and predictors of childfree people in developing countries

**Zachary P. Neal** *, **Jennifer Watling Neal**

Psychology Department, Michigan State University, East Lansing, Michigan, United States of America

* zpneal@msu.edu

**Data availability statement:** The Demographic and Health Survey (DHS) data use agreement prohibits sharing DHS data files. Instructions for requesting access to these data is

## Abstract

Childfree people—people who do not have children and do not want to have children in the future—represent and large and growing percentage of the population in wealthy countries. However, less is known about childfree people in developing countries. To facilitate this research, we developed software to identify childfree people in data from the Demographic and Health Surveys. Using this software, we estimated the prevalence of childfree people in 51 developing countries. Among single women ages 15–29, we found substantial cross-national and within-region variation in childfree prevalence, ranging from 0.3% in Liberia to 15.6% in Papua New Guinea. We also estimated the association between being childfree and country-level indicators of human development, gender equality, and political freedom. Results suggest that the prevalence of childfree people in a country is associated with the country's level of human development, and to a lesser extent their gender equality and political freedom. These results suggest that some developing countries have large populations of childfree people, and thus that being childfree is not a choice restricted to those living in the West or in wealthy countries. As developing countries evolve in terms of their human development, gender equality, and political freedom, it will be important to continue studying their childfree populations, both to understand demographic transitions in this part of the world, and to support its members' reproductive health and other needs.

## Introduction

A global decline in fertility rates [1,2] has led to an increased focus on people who do not have children [3–6]. *Childfree* people do not have children, but they also do not want to have children in the future. They have become the subject of books [7], documentaries [8,9], and magazine articles [10]. To date, the majority of research on childfree people has focused on wealthy countries in the Global North, where this population is large and growing [4–6,11–15]. However, less is known about this population in the developing world, where research has been restricted to married women [16,17] or to single countries and regions [17–19]. Although this prior work suggests that there are childfree populations in developing countries and thus lays the groundwork for the present study, it has provided limited insight into cross-national variations or the potential role of country-level characteristics such as levels of development.

available at https://dhsprogram.com/data/
Access-Instructions.cfm. The code necessary to
reproduce all results is available at
https://osf.io/9zjax/.

**Funding:** ZPN and JWN received funding from
the Michigan State University Asian Studies
Center (https://asia.isp.msu.edu/). The funder
played no role in the study design, data
collection, analysis, or preparation of the
manuscript.

**Competing interests:** The authors have no
competing interests to disclose.

In this study, we use data from the Demographic and Health Surveys (DHS) program that were collected between 2014 and 2023 on over 2 million people in 51 developing countries to pursue three research aims. First, we develop software to facilitate using DHS data, introducing the open-source `dhs()` function in the `childfree` package for R, which is now publicly available via the Comprehensive R Archive Network (CRAN). Second, we estimate and compare the country-level prevalence of childfree people in developing countries, finding significant variation in the percent of young single women who are childfree, ranging from a low of 0.3% in Liberia to a high of 15.6% in Papua New Guinea. Third, we explore country-level characteristics that are associated with a person being childfree, finding that human development has a strong positive association, while gender equality and political freedom have weaker associations.

The manuscript is structured in four sections. In the background section, we formally define 'childfree', describing our motivations for studying this population in developing countries and reviewing prior work in this context. In the methods section, we describe the DHS data, including how childfree respondents can be identified, and specify our plan for estimating prevalence and exploring predictors using these data. In the results section, we introduce newly-developed software for using these data, map comparative country-level prevalence estimates, and report the associations between being childfree and both individual-level and country-level characteristics. Finally, in the discussion section, we summarize our findings, noting their limitations and the directions they open for future research.

## Background

### Definitions

There is no single definition of 'childfree' (or 'voluntarily childless') that is widely accepted in the demographic literature. In this work, we adopt the definition from the Attitudes-Behaviors-Circumstances (ABC) framework for research on this population [3]. Specifically, we define a person as childfree if, at the time of data collection, they have not had any children and they do not want to have children in the future.

This definition distinguishes childfree people from several other of what the ABC framework calls 'family statuses.' Along the behavior dimension of the framework, it distinguishes childfree people from *parents* because childfree people have not had children, while parent have had children. Along the attitudes dimension of the framework, it also distinguishes childfree people from several other types of non-parents, including *not yet parents* who are planning to have children in the future, *undecided* people who are not sure if they want children. Finally, along the circumstances dimension, it distinguishes childfree people from *childless* people who wanted children but experienced biological or social barriers to having children.

The ABC definition of childfree has two important features that are worth noting. First, a person's status as childfree depends on their desire for children, and not on their fecundity. Thus, a non-parent is childfree if they do not want to have children, even if they are also biologically unable to have children. Second, a person's current status as childfree depends only on their current lack of children and current lack of desire for children in the future, and not on the possibility that later they may have or decide they want children. Thus, a person's status as childfree is potentially time-varying. Although studying people's trajectories through different family statuses over the life-course can also be valuable, in this cross-sectional study we are focused on whether people were childfree at the time of data collection.

## Childfree populations in developing countries

Despite decades of research in developing countries on fertility [20–24], which combines voluntary and involuntary childlessness into a single category and focuses broadly on whether or when people have children, quantitative demographic research on childfree people remains limited. An early study observed significant variation across 14 developing countries in the percentage of married women who were childfree in 1975, ranging from 0% in Lesotho and Sri Lanka to 4.8% in South Korea [16]. More recent studies have documented growth in the percentage of married women who are childfree in Iran (from 0.1% in 2000 to 0.3% in 2011) [17] and the Philippines (from 0.2-0.3% through 2013 to 0.5% in 2022) [18]. In the Philippines, this percentage was much higher among single women, but exhibited a similar rate of growth over the same period (from 4-5% through 2013 to 10.6% in 2022 [18]. The most comprehensive study to date focused on 38 countries in Sub-Saharan Africa, finding that 0.127% of women and 0.3% of men were childfree, and that being childfree was positively associated with three components of human development (education, health, income) [19].

This past research lays the foundation for a more detailed and comparative investigation of childfree populations in developing countries, while also raising two empirical questions. First, *in what developing countries are childfree people most and least common*? Country-level prevalence estimates are available for only 16 developing countries [16–18]. Moreover, because these estimates were computed at different times and using different operationalizations, they cannot be directly compared. Answering this question requires estimating childfree prevalence in a large sample of countries, during a single period, using a consistent operationalization.

Second, *what country-level characteristics are associated with being childfree*? Although there is some evidence that components of a country's level of human development are associated with being childfree [19], Second Demographic Transition (SDT) theory offers some additional possibilities [25,26]. SDT contends that subreplacement fertility and related demographic shifts (e.g., delayed marriage, decoupling of marriage and procreation) in highly developed countries, and increasingly in developing countries, are associated with macroscopic trends favoring the rise of the individual. One of these trends – rising human development – has previously been hypothesized [27] and demonstrated [19] to be associated with being childfree. Two other trends – "growth of solid democratic institutions" (p. 18113) and "cultural shifts toward more sex equality" (p. 18114) [25] – are hypothesized to have reduced fertility, which may occur in part because they increase the probability that a person will be childfree. Answering this question requires studying childfree populations drawn from a diverse sample of countries that vary on these dimensions.

In addition to helping answer these empirical questions, studying childfree people in developing countries can have practical significance for this population. Women in developing countries may experience pressure from partners and family members to become pregnant [28], and those who choose not to have children are frequently stigmatized [29]. Additionally, although choosing not to have children may offer women an opportunity to build wealth in countries with otherwise high poverty rates [30], it may also marginalize them from their families and other sources of economic support [31]. Finally, childfree people have specific reproductive health service needs (e.g., access to contraception, abortion, and voluntary sterilization) that are often unavailable [32–34]. Documenting the existence of childfree populations in developing countries may help to normalize the choice to be childfree, and may bring attention to this population's unmet needs.

### Research aims

Guided by this past research [16–19] and theory [25–27], the current study has three research aims. First, to facilitate this and future childfree research, we pursue the methodological aim of developing software designed to import, recode, and harmonize demographic data on childfree people (**Aim #1**). Second, we pursue the descriptive aim of estimating and comparing the country-level prevalence of childfree people in a large sample of developing countries (**Aim #2**). Third, we pursue the exploratory aim of identifying country-level characteristics that are associated with a person being childfree (**Aim #3**).

## Methods

### Data

The Demographic and Health Surveys (DHS) Program was a collaboration between the United States Agency for International Development (USAID) and local partners in developing countries to collect, harmonize, summarize, and disseminate population-representative data on a wide range of demographic and health indicators. At the time of writing, the DHS Program has been paused as a result of US Executive Order 14169, which suspended US foreign aid programs including those operated by USAID [35].

We focus on countries for which data were collected in the past 10 years from at least one of the following groups: single men 18–49, partnered men 18–49, single women 18–49, or partnered women 18–49. When multiple waves of these data were available from a single country, we use only the most recent wave. These data were accessed on 26 April 2023, and do not include information that could identify participants. This yielded an initial sample of 2,053,938 people in 52 countries. We excluded Timor-Leste ($N = 17,229$ people) as an outlier because the percentage of respondents who identified as childfree was more than 5 standard deviations above the mean. An additional 35,854 people were excluded because they were missing one or both of the variables used to determine whether they were childfree, and therefore their childfree status could not be determined. This yielded a final analytic sample of 2,000,855 people in 51 developing countries. The Demographic and Health Survey (DHS) data use agreement prohibits sharing DHS data files. Instructions for requesting access to these data is available at https://dhsprogram.com/data/Access-Instructions.cfm.

### Identifying childfree people

There are multiple ways to identify childfree people using commonly collected demographic and fertility preference data [3]. We use two variables available in the DHS to identify childfree people. First, `v201` (for women) or `mv201` (for men) measures the total number of children born to a respondent. The value of this variable is not obtained via a single survey question, but instead is computed based on a lengthy series of questions about the respondent's reproductive history. Second, `v613` (for women) or `mv613` (for men) captures respondents' numeric response to the question (in the local language): "*what is the ideal number of children you would like (or 'would have liked', if infecund) to have in your whole life, irrespective of the number of children you already have?*" A respondent is classified as childfree if they have not had any children (i.e., `(m)v201 = 0`) and they ideally do not want to have any children (i.e., `(m)v613 = 0`).

### Individual characteristics

Gender is measured using a binary indicator variable that is equal to 1 for women, and to 0 for men. marital status is measured using a binary indicator variable that is equal to 1 for

never married (i.e., always single) people, and equal to 0 for currently or formerly partnered people. Urbanicity is measured using a binary indicator variable that is equal to 1 for people living in an urban area, and equal to 0 for people living in a rural area. Finally, age is measured using a categorical variable that captures age quartiles: under age 22, age 22-29, age 30-39, and over age 39.

## Country characteristics

Past research [19] and theory [25] suggests that at least three country-level characteristics may be associated with whether a person is childfree: human development, gender equality, and political freedom.

We operationalize human development using the Human Development Index (HDI), which ranges from 0 (low) to 1 (high), and is a composite computed by the United Nations (UN) from life expectancy, education, and per capita income. We obtained HDI values for each country, corresponding to the year DHS data was collected in that country, from the UN Human Development Reports time series data [36]. Because 2023 HDI values were not available at the time of writing, we used 2022 values for Jordan and Senegal. HDI is correlated with GII at $r = -0.77$, and with GFS at $r = 0.43$.

We operationalize gender equality using the Gender Inequality Index (GII), which ranges from 0 (no inequality) to 1 (maximal inequality), and is a composite computed by the UN from indicators of three domains of opportunity cost to being a woman: reproductive health, empowerment, and labor market participation. We obtained GII values for each country, corresponding to the year DHS data was collected in that country, from the UN Human Development Reports time series data [36]. Because 2023 GII values were not available at the time of writing, we used 2022 values for Jordan and Senegal. Data on Chad is not available until 2018, so we used a linear interpolation of data from 2018–2022 to infer a value for 2014 when DHS data was collected in Chad. GII is correlated with HDI at $r = -0.77$, and with GFS at $r = -0.35$.

We operationalized political freedom using the Global Freedom Score (GFS), which ranges from 0 (no freedom) to 100 (maximally free), and is a composite computed by Freedom House from 10 political rights indicators and 15 civil liberties indicators. Prior to analysis, we divided GFS scores by 100 to place it on the same 0–1 scale as HDI and GII. We obtained GFS values for each country, corresponding to the year DHS data was collected in that country, from the Freedom in the World 2013–2024 data [37]. GFS is correlated with HDI at $r = 0.43$, and with GII at $r = -0.35$.

## Analysis plan

To achieve aim #1 – to develop software designed to import, recode, and harmonize demographic data on childfree people – we began with the existing software infrastructure provided by the `childfree` package for R [38]. This software already includes functions to use data from the U.S. National Survey of Family Growth (`nsfg()`) and the Michigan State of the State Survey (`soss()`), and provides an open-source template for also using data from the DHS.

To achieve aim #2 – to estimate and compare the country-level prevalence of childfree people – we computed within-country estimates of the percentage of people who are childfree using the R `survey` package with the sampling weights provided in the DHS data [39]. Because the universe of DHS samples varies across countries (e.g., only married women were surveyed in some countries), it is not possible to compute cross-nationally comparable prevalence estimates for complete country populations. Instead, we compute

cross-nationally comparable prevalence estimates for two subgroups included in all or most countries' sample universes. First, we estimate the percentage of ever-married women ages 17–49 who are childfree. This is the largest population subgroup for which a cross-nationally comparable estimate can be computed for all 51 countries. Second, we estimate the percentage of single women ages 15–29 who are childfree. Members of this population subgroup engage in particularly significant reproductive decision making, and cross-nationally comparable estimates can be computed for all but 5 countries (Afghanistan, Bangladesh, Egypt, Jordan, and Pakistan).

To achieve aim #3 – to identify country-level characteristics that are associated with a person being childfree – we estimate the association between country-level characteristics and people's childfree status, controlling for individual-level characteristics. To compute these estimates, we fit a a generalized linear mixed-effects model (GLMM) on the country-pooled data in which people are nested within their respective countries using the R lme4 package [40]. Following the practice of past studies using country-pooled DHS data [19], this model does not use sampling weights because these weights are solely a function of the individual-level demographic characteristics already included as covariates [41], and because there are no widely accepted or implemented methods for estimating weighted GLMMs.

The code necessary to reproduce these analyses and the results reported below is available at https://osf.io/9zjax/.

### Ethics statement

The DHS program approved access to restricted DHS Survey data for the purposes of researching "Prevalence of adults who do not want children" on 19 April 2023. The Michigan State University Institutional Review Board determined that analysis of these data does not constitute human subjects research on 25 April 2023 (STUDY00009134).

## Results

### Aim 1: The `childfree` R package

To process DHS data and to help others study childfree people using these data, we developed the dhs() function in the childfree package for R [38]. The function merges and harmonizes multiple DHS data files, identifies childfree respondents using the ABC framework [3], and recodes potentially useful demographic variables including sex, age, marital status, and urbanicity.

The package is distributed via the Comprehensive R Archive Network (CRAN), and can be installed using install.packages("childfree"). Once installed, the command dat <- dhs(files) will use the dhs() function to generate a harmonized dataframe called dat that combines data from multiple data files obtained from DHS. The package is accompanied by detailed documentation on its use (accessible using vignette("childfree")) a variable codebook (accessible using vignette("codebooks").

We used the newly-developed dhs() function to import, merge, and recode raw DHS data files. The analytic sample included 2,000,855 people across 51 countries (see Table 1), among whom 37,366 were childfree. Because the DHS is designed primarily as a survey to collect fertility data, some countries' samples are restricted to only women, or to only currently or formerly married people. As a result, a majority of the sample were currently or formerly married (72.75%) women (78.15%). However, the over-representation of married women in the total sample does not impact our prevalence estimates, which are restricted to specific

**Table 1**. Sample characteristics.

| Variable | Value |
| --- | --- |
| People | 2,000,855 |
| Childfree | 37,366 |
| Women | 78.15% |
| Never Married | 27.25% |
| Urban | 35.09% |
| Age under 22 | 24.37% |
| Age 22 - 29 | 25.36% |
| Age 30 - 39 | 27.14% |
| Age over 39 | 23.13% |
| Mean Age (SD) | 30.52 (10.33) |
|  |  |
| Countries | 51 |
| Mean HDI (SD) | 0.57 (0.11) |
| Mean GII (SD) | 0.53 (0.12) |
| Mean GFS (SD) | 0.45 (0.19) |

population subgroups, and does not impact our analysis of country-level characteristics, which includes gender and marital status as covariates. Reflecting the predominantly rural population distribution in developing countries, a minority of respondents lived in urban areas (35.09%). The sample had a mean age of 30.52 ($SD$ = 10.33), with respondents grouped into quartiles: under age 22 (24.37%), age 22-29 (25.36%), age 30-39 (27.14%), and over age 39 (23.13%).

The countries in which these people live were diverse with respect to their human development index ($M$ = 0.57, $SD$ = 0.11), gender inequality index ($M$ = 0.53, $SD$ = 0.12), and global freedom score ($M$ = 0.45, $SD$ = 0.19). Detailed country-level data are provided in the S1 Text, which is available at https://osf.io/9zjax/.

## Aim 2: Prevalence

Due to variations in each country's universe, the largest subgroup for which cross-nationally comparable prevalence estimates can be computed in all 51 countries was ever-married women ages 17–49. The prevalence of childfree people in this subgroup was small, ranging between 0% in Gambia to 0.9% in Papua New Guinea ($M$ = 0.0018, $SD$ = 0.0023), because, as we show below, marriage is strongly associated with a lower likelihood of being childfree.

Single women aged 15–29 are a potentially more interesting subgroup in the context of reproductive decision making, and cross-nationally comparable prevalence estimates can be computed for this subgroup in most countries. Fig 1 illustrates the percentage of single women ages 15–29 who are childfree in each country, by quartile; numeric estimates by country are provided in the S1 Text.

The average country-level prevalence among single women age 15-29 was 3.28% ($SD$ = 0.0355). However, mirroring a similar smaller analysis 50 years ago [16], there is substantial cross-national and within-region variation. For example, in Southeast Asia, prevalence is high in the Philippines (7.3%), but low in Indonesia (0.4%). In West Asia, prevalence is high in Turkey (5.3%), but low in Armenia (1%). In Sub-Saharan Africa, prevalence is high in Ethiopia (8.9%) and South Africa (5.7%), but low in Madagascar (0.4%) and Liberia (0.3%).

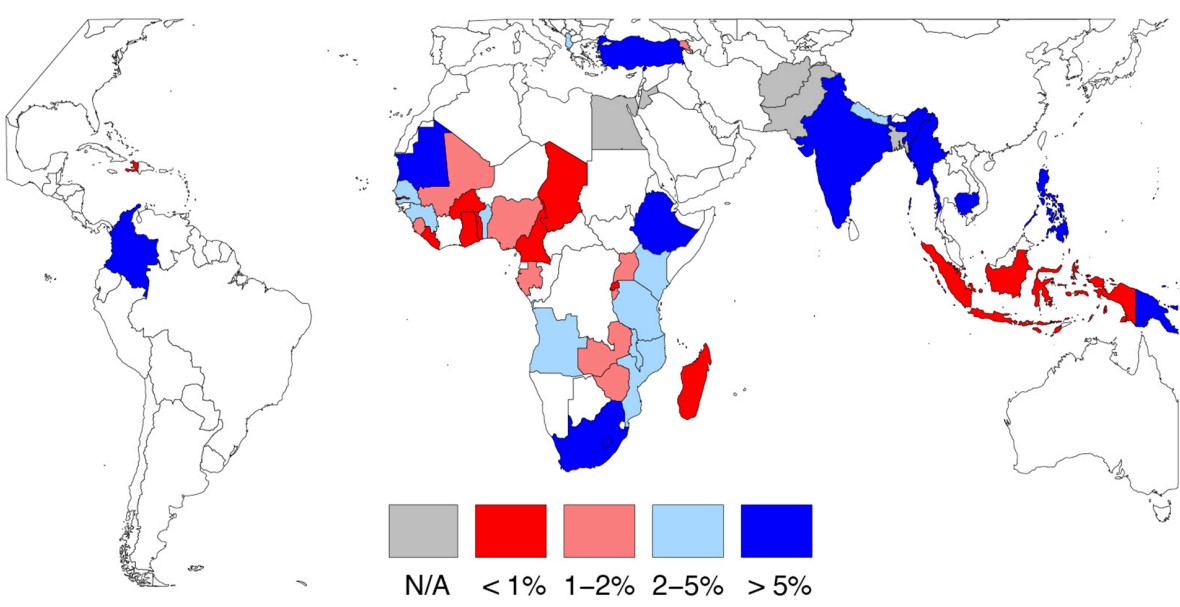

**Fig 1. Estimated percentage of single women age 15-29 who are childfree.**

## Aim 3: Predictors

The observed variation in prevalence of childfree status across countries, even within region, raises questions about the country-level characteristics that are associated with being childfree. Table 2 reports the results of a generalized linear mixed model predicting whether a person is childfree as a function of individual-level and country-level characteristics. Although the estimated within-country prevalence shown in Fig 1 focused only on single women ages 15–29, this model was estimated using the full sample of 2,000,855 people in 51 countries.

**Table 2. Estimates from a generalized linear mixed model (GLMM) predicting whether a person is childfree as a function of individual-level and country-level characteristics** ($N = 2,000,855$ **people in 51 countries**).

|  | B | SE | z | *p*-value | OR |
|---|---|---|---|---|---|
| Intercept | -9.336 | 0.068 | -137.672 | <0.001 | <0.001 |
|  |  |  |  |  |  |
| Individual-level Characteristics |  |  |  |  |  |
| Woman | -0.056 | 0.013 | -4.242 | <0.001 | 0.946 |
| Never Married | 3.631 | 0.021 | 172.886 | <0.001 | 37.740 |
| Urban | -0.05 | 0.012 | -4.248 | <0.001 | 0.951 |
| Under age 22 | omitted category |  |  |  |  |
| Age 22-29 | -0.187 | 0.014 | -13.637 | <0.001 | 0.829 |
| Age 30-39 | 0.148 | 0.020 | 7.424 | <0.001 | 1.160 |
| Over age 39 | 0.595 | 0.024 | 24.9 | <0.001 | 1.812 |
|  |  |  |  |  |  |
| Country-level Characteristics |  |  |  |  |  |
| HDI | 3.913 | 0.047 | 82.633 | <0.001 | 50.027 |
| GII | 0.486 | 0.054 | 8.952 | <0.001 | 1.625 |
| GFS | -0.700 | 0.062 | -11.266 | <0.001 | 0.496 |

The individual-level characteristics are not of direct interest for research aim #3, but were included to isolate the independent association of childfree status with the country-level characteristics. Women were slightly less likely to be childfree than men ($B = -0.056$, $SE = 0.013$, $p<0.001$, $OR = 0.946$), and people living in urban areas were slightly less likely to be childfree than people living in rural areas ($B = -0.05$, $SE = 0.012$, $p<0.001$, $OR = 0.951$). Never married (i.e., single) people were much more likely to be childfree than currently or formerly married people ($B = 3.631$, $SE = 0.021$, $p<0.001$, $OR = 37.740$). Finally, compared to people under age 22, those who were age 22-29 were less likely to be childfree ($B = -0.187$, $SE = 0.014$, $p<0.001$, $OR = 0.829$), while those who were age 30-39 ($B = 0.148$, $SE = 0.020$, $p<0.001$, $OR = 1.160$) and over age 39 ($B = 0.595$, $SE = 0.024$, $p<0.001$, $OR = 1.812$) were progressively more likely to be childfree. The strength of the associations between childfree status and these individual-level demographic characteristics was generally small. Marital status is an exception and had a strong association with childfree status, which is consistent with prior studies in the United States [12], Japan [13], and the Philippines [18].

At the country level, all three characteristics of interest were associated with being childfree. First, people living in countries with higher levels of human development (HDI) were more likely to be childfree ($B = 3.913$, $SE = 0.047$, $p<0.001$, $OR = 50.027$). Second, people living in countries with higher levels of gender inequality (GII) were more likely to be childfree ($B = 0.486$, $SE = 0.054$, $p<0.001$, $OR = 1.625$). Third, people living in countries with greater political freedom (GFS) were less likely to be childfree ($B = -0.700$, $SE = 0.062$, $p<0.001$, $OR = 0.496$).

Because the sample is large, all of the coefficients are statistically significant, even if their estimated associations are small. To clarify the practical significance of the associations between country-level characteristics and childfree status, Fig 2 plots the estimated probability (with 99% confidence interval) that a single woman age 22-29 is childfree as a function of each characteristic, holding the other two characteristics constant at the sample's median. Fig 2A shows that variation in countries' human development were associated with large differences in the probability of their residents being childfree, with prevalence rates among young women in low-development countries such as Chad (HDI = 0.388) around 1%, but rates in high-development countries such as Turkey (HDI = 0.838) around 6%. In contrast, Fig 2B and 2C show that variation in countries' levels of gender inequality and freedom were associated with relatively small differences in the probability of their residents being childfree.

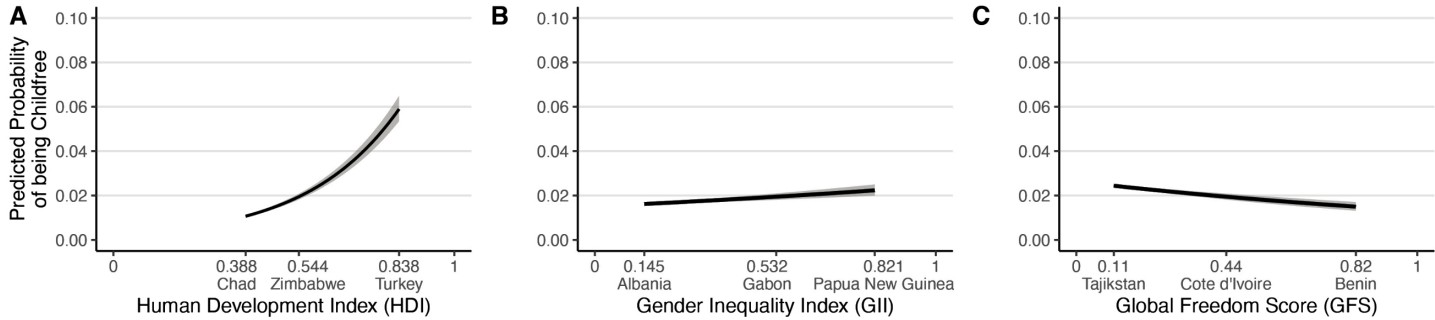

**Fig 2. Estimated probability (with 99% confidence interval) that a single woman age 22-29 is childfree as a function of their country's (A) human development index, (B) gender inequality index, and (C) global freedom score, holding the other two country-level characteristics constant at the median.** For context, the x-axis labels the minimum, maximum, and median country and value of the respective characteristic in this sample.

## Discussion

Declining global fertility rates have raised demographers' and policymakers' interest in people who do not have children, and specifically in those who do not want to have children in the future and therefore are *childfree*. Most research on childfree people has been conducted in wealthy countries, while less is known about this population in developing countries, where the few quantitative demographic studies have been restricted to married women [16,17] or specific regions [17–19]. Guided by this prior work and the predictions of Second Demographic Transition (SDT) theory, we used data from the Demographic and Health Surveys program on over 2 million people in 51 developing countries to understand where they are located, and what country-level characteristics are associated with being childfree.

Our first research aim was to develop software to facilitate research on childfree people in developing countries. To this end, we created, documented, and distributed the `dhs()` function as part of the `childfree` package for R [38]. This function enables researchers to construct custom demographic data from raw DHS data files that identify childfree respondents and contains recoded and harmonized demographic variables. It is significant because it simplifies not only research on childfree people in the developing world, but also cross-national and longitudinal research on this population.

Our second research aim was to estimate and compare the country-level prevalence of childfree people. Because variations in DHS samples across countries mean that a whole-population prevalence cannot be estimated for many countries, we instead estimated cross-nationally comparable prevalences for two subgroups: ever-married women age 17-49 and single women age 15-29. On average, only 0.18% of a country's ever-married women age 17-49 are childfree, while 3.28% of a country's single women age 15-29 are childfree. However, these averages mask substantial cross-national and intra-regional variations. For example, many single young women are childfree in the Philippines (7.3%), while few are childfree in Indonesia (0.4%). These findings are significant for two reasons. First, these are the first cross-nationally comparable estimates of childfree prevalence in the 21$^{st}$ century, and provide a benchmark against which changes in prevalence can be evaluated. Second, they highlight that some developing countries have large populations of childfree people, and that childfree people are as common in some developing countries as they are in developed countries. Thus, they suggest that being childfree is not a choice restricted to those living in the West or in wealthy countries.

Our third research aim was to identify country-level characteristics that are associated with being childfree. Guided by SDT, we focused on the potential associations of human development, gender equality, and political freedom. Using a generalized linear mixed model to pool all respondents from all countries, controlling for their individual demographic characteristics, we found that a country's human development index had a statistically significant and strong positive association with its residents being childfree. This is consistent with long-standing predictions about the role of human development in suppressing fertility [19,25,27]. In contrast, a country's gender inequality index and global freedom score had statistically significant but relatively weak associations. Although these forces are referenced by SDT [25], the theory has been critiqued for is ambiguity on their roles and more broadly for its treatment of gender and globalization [26]. These findings are significant because they extend support for the role of human development on reproduction, from its previously documented effects on fertility (i.e., women have *fewer children* on average), to its effects on being childfree (i.e., more people do not want to have *any children*).

These results must be interpreted in light of some limitations, which identify directions for future research. First, DHS data are cross-sectional and in this study we adopt a compatible

definition of childfree as referring to a respondent's current lack of children and their current lack of desire for children in the future [3]. This is not a limitation in itself, but it does mean than we are unable to identify respondents who are not childfree but later become childfree (e.g., undecided → childfree), or respondents who are childfree but later have children (i.e., childfree → parent). Future research should explore opportunities to use panel data to examine people's transitions into and out of a childfree status over the life course. Second, as an exploratory study in a context with limited prior quantitative research, we focused on a limited set of country-level characteristics selected based on SDT. The `dhs()` function that we developed enables future research to use these data to explore other potentially relevant country-level characteristics including the prevalence of specific health conditions (e.g., maternal mortality, HIV), cultural forces (e.g., religion), and climate impacts. Finally, alongside all other USAID-administered programs, the DHS program has been paused indefinitely, which means that access to existing and future data may be limited. All three of these limitations are related to deficiencies in the collection and availability of demographic data in developing countries and on childfree people. Although they are solvable in principle, for example through small-scale panel modules, these solutions would likely require a national or international program like the DHS to achieve [35].

Despite these limitations, this study sheds new light on childfree people in developing countries. The large size and rapid growth of the childfree population in developed countries is well-documented [4], but this work suggests there are also large childfree populations in some developing countries. Moreover, we find that the size of these populations is likely related to the countries' level of human development, and to a lesser extent their gender equality and political freedom. As these countries continue developing along each of these dimensions, it will be important to continue studying their childfree populations, both to understand demographic transitions in this part of the world, and to support its members' reproductive health and other needs.

## Supporting information

**S1 Text. Country characteristics and estimated childfree prevalence.**
(PDF)

## Author contributions

**Conceptualization:** Zachary P. Neal, Jennifer Watling Neal.

**Data curation:** Zachary P. Neal, Jennifer Watling Neal.

**Formal analysis:** Zachary P. Neal, Jennifer Watling Neal.

**Funding acquisition:** Zachary P. Neal, Jennifer Watling Neal.

**Investigation:** Zachary P. Neal, Jennifer Watling Neal.

**Methodology:** Zachary P. Neal, Jennifer Watling Neal.

**Software:** Zachary P. Neal, Jennifer Watling Neal.

**Visualization:** Zachary P. Neal, Jennifer Watling Neal.

**Writing – original draft:** Zachary P. Neal, Jennifer Watling Neal.

**Writing – review & editing:** Zachary P. Neal, Jennifer Watling Neal.

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
