## [Decision Letter · Decision Letter 0]

21 Jul 2025

PONE-D-25-19934Prevalence and predictors of childfree people in developing countriesPLOS ONE

Dear Dr. Neal,

Thank you for submitting your manuscript to PLOS ONE. After careful consideration, we feel that it has merit but does not fully meet PLOS ONE’s publication criteria as it currently stands. Therefore, we invite you to submit a revised version of the manuscript that addresses the points raised during the review process.

Three experts have reviewed the manuscript noting that there are substantial revisions required including the literature review, the key defininitons in the methodology, the analysis and the reporting of results.

We look forward to receiving your revised manuscript.

Kind regards,

José Antonio Ortega, Ph.D.

Academic Editor

PLOS ONE

Journal Requirements:

“This research received funding from the Michigan State University Asian Studies Center. 323”

“ZPN and JWN received funding from the Michigan State University Asian Studies Center (https://asia.isp.msu.edu/). The funder played no role in the study design, data collection, analysis, or preparation of the manuscript.”

Reviewers' comments:

Reviewer's Responses to Questions

**Comments to the Author**

1. Is the manuscript technically sound, and do the data support the conclusions?

Reviewer #1: Yes

Reviewer #2: Yes

Reviewer #3: Partly

2. Has the statistical analysis been performed appropriately and rigorously? 

Reviewer #1: Yes

Reviewer #2: Yes

Reviewer #3: No

3. Have the authors made all data underlying the findings in their manuscript fully available?

Reviewer #1: Yes

Reviewer #2: Yes

Reviewer #3: Yes

4. Is the manuscript presented in an intelligible fashion and written in standard English?

Reviewer #1: Yes

Reviewer #2: Yes

Reviewer #3: Yes

5. Review Comments to the Author

Reviewer #1: Thank you for the opportunity to review this paper on the prevalence and predictors of childfree individuals in developing countries. Given the global decline in fertility rates, this study addresses an important and under-researched topic. I have several suggestions for the authors to consider:

1. One of the motivations for examining factors associated with childfree individuals in developed countries is to better understand declining fertility rates and inform pro-natalist policy responses. Could the authors provide more context on fertility trends in the developing countries included in this study? Are these countries also experiencing declining fertility?

2. I recommend that the authors include the original survey questions used to identify childfree individuals in this study.

3. It is unclear why gender interaction analyses were conducted only for country-level factors, and not for individual-level factors. Given the likelihood that individual-level predictors may have differential effects by gender, the authors should consider testing gender interactions for these variables as well.

4. Age is currently dichotomised as under 30 versus 30 or older. If the data permit, a more detailed categorisation (e.g. 18–29, 30–39, and 40–49) would allow for a more nuanced interpretation of age-related patterns.

5. The authors should reconsider how country-level variables are modelled and interpreted. Including them as continuous variables ranging from 0 to 1 means that the reported odds ratios represent comparisons between theoretical extremes (e.g. 0 vs. 1), which may be unrealistic. Rescaling these variables to a 0–100 scale or treating them as categorical may enhance interpretability.

6. The discussion would benefit from a comparison of the findings with those from similar studies conducted in developed countries.

7. In the results, the authors mentioned ‘due to the design and sampling strategy used by the DHS, the majority of these individuals were currently or formerly married.’ It would be helpful to provide more detail on the DHS sampling strategy and discuss how this may affect the generalisability of the findings.

8. I was unable to locate the supplementary materials referenced in the manuscript.

Reviewer #2: Thank you for the opportunity to review this manuscript. It was a pleasure to read and provides an important perspective on an emerging topic in LMIC. My primary comments are in order to clarify the communication in a few of the sections of the manuscript.

-Line 46. The last sentence of this paragraph doesn't clarify the purpose of the focus on countries' human development, gender inequality, and political freedoms. We can posit that the purpose is to focus on these aspects in relation to the levels of childfree individuals, but this needs to be clarified.

-Definition of childfree. If you are using the definition of childfree as described, there are a few things that need to be discussed either in the methods section or in the limitations. Both number of children born and number of ideal children are time varying and could be influencing each other in either direction. For example, if a woman (or man) is infertile and is not able to have children, that inability to have children may influence the ideal number of children that a person would report. The persons report may be biased to reflect the status quo. I don't think this needs to change your definition, but is worth mentioning in the limitations.

-Methods section. I would appreciate some clarification in this section. I believe that the within-country estimates were calculated by country, but the multilevel regression was estimated on a dataset with all the countries combined, but this is not described anywhere. I would want to know how the individuals in that pooled dataset were weighted as well.

-Interactions. The cross-level interactions are included in Table 2, but is not discussed anywhere in the paper, in the results or discussion. It would be useful to present these along with their interpretation.

Great job on this very interesting paper.

Reviewer #3: GENERAL COMMENT

Using DHS data from 51 countries and more than 2 million respondents, the authors estimate that around 1.9% of people are childfree and explore individual and country-level predictors through multilevel modeling. This is potentially valuable work, especially because childlessness, regardless of being voluntarily or not, has important implications for fertility decline, population aging, and policy responses.

Despite this potential, the manuscript falls short in several ways. The theoretical framing is weak, some definitions are incorrect, the methods lack transparency, and many claims are overstated given the data. Most importantly, the paper does not leave the reader with a clear take-home message. After finishing the paper, it is hard to answer the key question: what do these findings really tell us about demographic change in developing countries? The paper could make a meaningful contribution, but only if the authors substantially revise and restructure it to focus on a clear story.

CONTRIBUTION & FRAMING

1. The authors claim that research on childfree people in developing countries has been "largely neglected." This is a bold claim that should be softened. Voluntary childlessness has been studied before in developing contexts, though several key studies are missing from the literature review. Such as:

- De Talancé, M. (2019). Education, fertility and childlessness in Indonesia.

- Poston, D. L., Kramer, K. B., Trent, K., & Yu, M. Y. (1983). Estimating voluntary and involuntary childlessness in the developing countries. Journal of Biosocial Science, 15(4), 441-452.

- Bagi, M. (2023). Prevalence, reasons and consequences of childlessness in the world and Iran: a systematic review. Journal of Population Association of Iran, 18(35), 97-148.

- Ibisomi, L., & Mudege, N. N. (2014). Childlessness in Nigeria: perceptions and acceptability. Culture, Health & Sexuality, 16(1), 61-75.

2. In light of point 1, the real novelty of this paper is rather in the cross-national DHS-based estimates and predictors, and the authors should say so clearly.

3. The biggest weakness is that the paper does not clearly explain why these findings matter. What does a 1.9% prevalence mean for population trends? Does this indicate early signs of the Second Demographic Transition spreading to the Global South? Or is voluntary childlessness still too rare to have demographic consequences? What are the main predictors? What does it mean for policy or future implication of demographic trends? A strong revision should answer these questions directly to allow a clear take-home message of the study.

4. I was surprised by the lack of theoretical grounding. The introduction frames the study around the "unique needs" of childfree people, rather than focusing on individual- or population-level implications of childlessness.

5. The fact that there is no explicit theoretical framework also tone down the choice of macro-level predictors, which seems rather arbitrary. The omission of Second Demographic Transition (SDT) theory is particularly notable. SDT offers a clear rationale for expecting links between development, gender equality, political freedoms, and voluntary childlessness, as it emphasizes individualization, gender equity, and post-materialist values. The lack of theoretical grounding makes the selection of HDI, GII, and GFS appear ad hoc.

6. The authors repeatedly define childfree individuals as those who "do not have or want children." The standard demographic definition is "do not have AND do not want children." This should be corrected everywhere.

DATA AND MEHTODS

7. The choice of DHS data raises significant concerns. While DHS is valuable for fertility research, it is not ideally suited to studying voluntary childlessness. The paper does not discuss this limitation adequately. DHS samples differ across countries—some include only married women—making cross-national comparability problematic. The authors acknowledge this briefly but proceed to make sweeping cross-country statements.

8. Given point 7, previous studies who make use of DHS-based estimates for childlessness in developing countries (such as Chua et al. 2025) or other studies that use different data should be used as benchmark. For example: Chua et al. 2025

9. The operationalization of childfree status is also problematic. Respondents reporting 0 ideal children are classified as childfree, but ideal fertility preferences, particularly among young people, often change under social pressure. The authors briefly acknowledge that DHS is cross-sectional but do not explain why this is a critical limitation for interpreting childfree as a life-course outcome.

10. Overall, the methodological details given to the reader are insufficient to ensure reproducibility. The supplementary information reference in the text is unavailable for review, and no detailed country-level table with sample size, survey year, percentage of childlessness in country-year is given. Some comments are below:

- The time frame of the data is ambiguous. The authors state they used the "most recent DHS from the past 10 years," but it is unclear whether data were pooled across multiple rounds or restricted to one survey per country. This ambiguity undermines claims about temporal trends, such as an "emerging childfree population," which cannot be inferred from cross-sectional data.

- 35,854 respondents with "undetermined" status were excluded without explanation. Were they disproportionately young, unmarried, or missing key variables?

- As far as I understand, infecund individuals are recoded in data, but not clearly excluded from the analyses. This conflates voluntary and involuntary childlessness.

- The sample is strongly unbalanced with 78% of respondents being women, yet no justification is given.

- The model omits key individual-level predictors such as education, wealth, and urban/rural residence, all known to correlate with voluntary childlessness.

- Age is modeled very crudely as a binary variable (<30 vs ≥30), which ignores well-known life-course patterns in fertility decision making. Demographic research typically models age continuously or in 5-year reproductive cohorts to distinguish temporary postponement among younger people from permanent childfree status at older ages. The authors should consider using finer age groups or, at minimum, present age-specific prevalence for older cohorts (e.g., 40-49) to better reflect permanent voluntary childlessness.

- HDI, GII, and GFS are highly correlated, yet no multicollinearity diagnostics (e.g., VIFs) are presented. Other relevant indicators, such as maternal mortality, are discussed but ignored in the models. Are they available? If yes, why are they excluded? (this also recalls the need of a theory or a research questions guiding the analyses)

RESULTS/INTERPRETATION

Several findings appear implausible or are overinterpreted:

11. The reported childfree prevalence cannot be generalized to entire populations because the DHS sample is heavily skewed toward women and married individuals. The authors should emphasize this limitation. For instance, Indonesia's DHS-based childfree prevalence (0.4%) is higher than its reported permanent childlessness (0.22%) in the Indonesian Family Life Survey (De Talancé 2019). By definition, voluntary childlessness should not exceed total childlessness. The authors should conduct plausibility checks against external sources and discuss possible reasons for these discrepancies.

12. The authors report that almost half of the total sample is under age 30 and that childfree prevalence is highest among younger women (15-29 years). Figure 1 explicitly focuses on single young women, where prevalence reaches over 5% in some countries. This suggests that childfree status is disproportionately concentrated among younger individuals in the DHS sample. However, the multilevel model reports: "Those under age 30 (B = -0.271, OR = 0.762, p<0.001) were less likely to be childfree." This means that, after controlling for other factors, younger people are less likely to be childfree than older people. This discrepancy likely reflects conceptual and methodological issues: (1) the classification of young respondents with 0 ideal children as childfree, even though many may later have children; (2) strong confounding by marital status, which reverses the age effect once controlled for; and (3) the skewed sample composition. The authors should clarify this contradiction, ideally by providing age-specific prevalence and discussing the life-course dimension of voluntary childlessness.

13. The explanation offered for the fact that older people are more likely to be childfree - that younger people "receive messages about the importance of having children" - is vague and unconvincing. A life-course explanation (temporary postponement vs permanent childlessness) would be more appropriate.

14. Many of the reported associations, particularly for HDI, GII, and GFS, are statistically significant but substantively trivial given the very large sample size. For example, gender differences in childfree prevalence at high HDI levels (Figure 2A) and slopes for gender inequality (Figure 2B) and political freedom (Figure 2C) are extremely small and should be interpreted cautiously. Moreover, confidence intervals are missing from the figures, making it impossible to judge the reliability of these estimates. The authors should include uncertainty intervals, present predicted probabilities, and tone down claims of meaningful macro-level effects.

15. The paper fails to emphasize that DHS only captures current fertility preferences, not stable life-course decisions. Statements about an "emerging" childfree population are unjustified.

MINOR CONCERNS

16. Be aware that there are multiple typographical errors (e.g., "its' members") and informal phrases ("want questions") that should be corrected before a second submission.

6. PLOS authors have the option to publish the peer review history of their article (what does this mean?). If published, this will include your full peer review and any attached files.

Reviewer #1: No

Reviewer #2: No

Reviewer #3: No

---

## [Decision Letter · Decision Letter 1]

21 Sep 2025

Prevalence and predictors of childfree people in developing countries

PONE-D-25-19934R1

Dear Dr. Neal,

We’re pleased to inform you that your manuscript has been judged scientifically suitable for publication and will be formally accepted for publication once it meets all outstanding technical requirements.

Kind regards,

José Antonio Ortega, Ph.D.

Academic Editor

PLOS ONE

Additional Editor Comments (optional):

The extensive revision has made clearer the purpose of the paper and its contribution. Note that, given stated aim 1 on software, the manuscript must satisfy requirements for such papers in PLOS ONE, namely utility, validation and availability. In the opinion of the editor the three requirements are satisfied: it provides added functionality to an already existing package and it is available in the revised R package from CRAN. The effort to provide estimates for most DHS countries is an example of validation, and the authors make a good effort in comparing results from surveys with different population universes.

Two of the previous referees were available. #1 suggests acceptance, #2 raises two minor issues: that the reasons for not providing some estimates are not incorporated. I think they are, and they can be found in the supplementary material. The reason to focus on single women 15-29 as one of the two populations under comparison also is justified in the text. Regarding the request to include trends, I think the purpose of providing one estimate per country is stated by the authors and it is their right to do so. Reviewer #3 was not available but the issues raised have been adequately dealt with in the opinion of the editor.

Reviewer #1:

Reviewer #2:

Reviewers' comments:

Reviewer's Responses to Questions

**Comments to the Author**

1. If the authors have adequately addressed your comments raised in a previous round of review and you feel that this manuscript is now acceptable for publication, you may indicate that here to bypass the “Comments to the Author” section, enter your conflict of interest statement in the “Confidential to Editor” section, and submit your "Accept" recommendation.

Reviewer #1: All comments have been addressed

Reviewer #2: (No Response)

2. Is the manuscript technically sound, and do the data support the conclusions?

Reviewer #1: Yes

Reviewer #2: Yes

3. Has the statistical analysis been performed appropriately and rigorously? 

Reviewer #1: Yes

Reviewer #2: Yes

4. Have the authors made all data underlying the findings in their manuscript fully available?

Reviewer #1: Yes

Reviewer #2: Yes

5. Is the manuscript presented in an intelligible fashion and written in standard English?

Reviewer #1: Yes

Reviewer #2: Yes

6. Review Comments to the Author

Reviewer #1: The authors have adequately addressed all of my previous comments. I have no further suggestions or comments.

Reviewer #2: Very nice revision to your original manuscript. I believe that it is acceptable for publication, with a few suggestions on making the analysis more clear and one suggestion to your limitations/areas for future research.

You highlight the prevalence of childfree people in the group of single women ages 15-29, but have very limited justification for doing so, simply stating that they make significant reproductive decisions. I would encourage you to more fully describe your reasons for choosing this group. You also state that you could not estimate the prevalence of childfree people in this group in 5 countries, but neglect to say why.

In addition, in the aim 3 analysis, you find that as individuals get older (after age 30), they are more likely to be childfree compared to those under age 22. I was wondering if from earlier studies on earlier DHS surveys using the same definition of childfree you would be able to in effect look at changes in the estimate for an age cohort to see if there were any changes in the prevalence of childfree individuals within a country over time. This could give an idea of whether the childfree status is remaining stable or is fluctuating over time.

7. PLOS authors have the option to publish the peer review history of their article (what does this mean?). If published, this will include your full peer review and any attached files.

Reviewer #1: **Yes: **Chuyao Jin

Reviewer #2: No

---

## [Editor Report · Acceptance letter]

PONE-D-25-19934R1

PLOS ONE

Dear Dr. Neal,

I'm pleased to inform you that your manuscript has been deemed suitable for publication in PLOS ONE. Congratulations! Your manuscript is now being handed over to our production team.

Kind regards,

on behalf of

Dr. José Antonio Ortega

Academic Editor

PLOS ONE